# *Gleditsia triacanthos* Galactomannans in Gluten-Free Formulation: Batter Rheology and Bread Quality

**DOI:** 10.3390/foods12040756

**Published:** 2023-02-09

**Authors:** Lorena S. Sciarini, Pablo M. Palavecino, Pablo D. Ribotta, Gabriela N. Barrera

**Affiliations:** 1Instituto de Ciencia y Tecnología de los Alimentos Córdoba (ICYTAC-CONICET), Universidad Nacional de Córdoba (UNC), Av. Filloy s/n, Córdoba CP X5000HUA, Argentina; 2Facultad de Ciencias Agropecuarias (FCA), Universidad Nacional de Córdoba (UNC), Av. Valparaíso s/n, Córdoba CP X5000HUA, Argentina; 3Facultad de Ciencias Exactas, Físicas y Naturales (FCEFyN), Universidad Nacional de Córdoba (UNC), Av. Vélez Sarsfield, 1611, Córdoba CP X5000HUA, Argentina

**Keywords:** *Gleditsia triacanthos*, galactomannan, gluten-free, bread

## Abstract

Gluten-free batters, in general, require the incorporation of agents to control their rheology; this role is commonly played by hydrocolloids. New natural sources of hydrocolloids are under permanent research. In this regard, the functional properties of the galactomannan extracted from the seed of *Gleditsia triacanthos* (Gledi) have been studied. In this work, we evaluated the incorporation of this hydrocolloid, alone and in combination with Xanthan gum, in gluten-free batters and bread and compared it with Guar gum. The incorporation of hydrocolloids increased the viscoelastic profile of the batters. Gledi addition at 0.5% and 1.25% increased the elastic modulus (G′) by 200% and 1500%, respectively, and similar trends were observed when Gledi-Xanthan was used. These increases were more pronounced when Guar and Guar-Xanthan were used. The batters became firmer and more elastically resistant because of the addition of hydrocolloids; batters containing Gledi had lower values of these parameters than batters containing Gledi-Xanthan. The addition of Gledi at both doses significantly increased the volume of the bread compared to the control by about 12%, while when Xanthan gum was included, a decrease was observed, especially at higher doses (by about 12%). The increase in specific volume was accompanied by a decrease in initial crumb firmness and chewiness, and during storage, they were significantly reduced. Bread prepared with Guar gum and Guar-Xanthan gum combinations was also evaluated, and the trends observed were comparable to that of bread with Gledi gum and Gledi-Xanthan gum. The results showed that Gledi addition favors the production of bread of high technological quality.

## 1. Introduction

Celiac disease (CD) is an autoimmune disorder caused by the consumption of gluten proteins; the only treatment currently available for this disease is patient adherence to a lifelong gluten-free diet. The frequency of CD has been reported to double every two decades [1]. This, coupled with the perception of some non-celiac consumers that the gluten-free diet is healthier, has resulted in a large increase in the size of the market for these food products. Thus, some reports estimate the size of the gluten-free market to be USD 5.7 billion in 2020, and it is projected to reach USD 8.3 billion in 2025 [2]. This situation poses a challenge for the scientific community and the food industry. Finding new ingredients and combinations of raw materials and production processes to obtain gluten-free food is necessary to fulfill consumers’ demand for this type of product.

Among gluten-free products, the manufacture of quality gluten-free bread is one of the most challenging tasks, largely due to the lack of alternative ingredients that can mimic wheat protein functionality [3]. The formulation of gluten-free bread mainly involves starch, protein-based ingredients (soy products, dairy ingredients, egg) and hydrocolloids (as additives) into a gluten-free base flour. These ingredients present a high-water absorption capacity, and thus gluten-free formulations need higher water amounts than wheat-bread formulations. Therefore, bread dough is transformed into a batter. Water level and distribution influence the rheological behavior of gluten-free batter and, consequently, the performance of the baked-good products. Protein-based ingredients and hydrocolloids help to improve the structural properties of the gluten-free matrix, and the formers also contribute to improving the nutritional profile of bread, increasing its protein content [4]. Hydrocolloids reinforce the batter matrix by encouraging extensive structuring of the surrounding water of the aqueous system, which leads to batters-stabilization by preventing settling, phase separation and foam collapse. In gluten-free batters, gas retention is mainly controlled by the system’s viscosity since batters lack elastic properties due to the absence of a protein network. Water-binding components such as hydrocolloids enhance the viscoelastic behavior of batters by increasing viscosity, which improves batter expansion during the fermentation process by stabilizing gas cells [5].

The most used hydrocolloids in gluten-free breadmaking are Xanthan gum and Hydroxypropylmethyl cellulose [6]. However, nowadays, consumers prefer ingredients that are more natural, and they consider these hydrocolloids as artificial [7]. In this regard, hydrocolloids from seeds are potential alternatives for the manufacture of gluten-free bread [8] while being perceived as more natural. Examples of these hydrocolloids are Guar (*Cyamopsis tetragonolobus*; Mannose:galactose, Man:Gal, ratio 2:1), Locust bean (*Ceratonia siliqua*; Man:Gal ratio 4:1), and Tara (*Caesalpinia spinosa*; Man:Gal ratio 3:1) gums.

The effect of Guar and Locust bean gum, among other hydrocolloids, on gluten-free bread quality was evaluated by Acs et al. [9], and they demonstrated that the incorporation of these galactomannans results in volume improvements and crumb softening. Casas et al. [10] reported a synergistic interaction between Xanthan and Guar gums, which resulted in increased viscosities. In this regard, the combinations of Xanthan-Guar gum and Xanthan-Locust bean gum have been demonstrated to be effective in improving dough structure [11]. Although these galactomannans are commercially important in the food industry since they have several functions, including as moisturizers, thickeners, stabilizers, and binders, innovative or non-traditional sources of gums are currently being investigated in order to explore new products with similar or better functionalities and/or new applications. 

Concerning the exploration of non-traditional sources of gums, the physicochemical and rheological properties of plant seed-derived galactomannans have been studied in the last years [12,13,14,15,16,17,18,19]. *Gleditsia triacanthos* (Fabaceae; Gt) is a woody species that blooms in the southern hemisphere and is widely spread in Argentina [20], despite being native to North America and Central Europe [21]. In a previous study, the improving effect on the film-forming properties of soy protein was reported when the hot-water soluble galactomannan extracted from Gt seeds was used as a co-component [22]. It was found that improved films could be obtained through the combination of both components in certain ratios in relation to single-component samples. The galactomannan extracted from Gt seeds with boiling water could be a potential additive in the food industry [23,24] and be used as an alternative to Guar and Locust bean gums [25]; moreover, synergistic interaction between this and other hydrocolloids has been reported [26,27,28,29]. However, it is important to mention that the extract of *Gleditisia triacanthos* has not been admitted yet as a food additive in Argentine legislation.

No reports were found on the use of Gt polysaccharides as an ingredient in gluten-free bread formulations, so this study is the first to analyze the effect of *Gleditsia triacanthos* galactomannan’s extract on batter-rheology and bread-making performance in a gluten-free formulation.

## 2. Materials and Methods

### 2.1. Materials

Commercial rice flour (Glutal, Santa Fe, Argentina), cassava starch (Dimax, Córdoba, Argentina) and full-fat active soy flour (NICCO, Córdoba, Argentina), compressed yeast (Calsa, Buenos Aires, Argentina), shortening (Dánica, Buenos Aires, Argentina) and salt (Dos Anclas, Buenos Aires, Argentina) were used for gluten-free bread formulation.

*Gleditsia triacanthos* pods were manually collected from trees in Córdoba (center and northwest region of the province), Argentina. The seeds were mechanically separated from the pods and manually cleaned and classified. The seeds were milled in a hammer grinder (Pulverisette 16, Fritsch, Idar-Oberstein, Germany) [26]. Additionally, commercial Guar and Xanthan gums (Saporiti, Buenos Aires, Argentina) were used as controls to compare the galactomannans fraction extracted from Gt seeds in gluten-free bread formulation, considering that both gums are among the most popular hydrocolloids used in these products. 

### 2.2. Galactomannan Extraction from Milled Seeds of Gt

Galactomannans were extracted from Gt-milled seeds with hot water and ethanolic precipitation, according to Sciarini et al. (2009). The precipitated material was dried in a hot, dry oven (Memmert 600, Schwabach, Germany) at 35 ± 3 °C for 10 h (% Yield = 15.3) and was then milled in a cyclonic grinder (Ciclotec™ Hillerød, Denmark). The yellowish-powder extract (galactomannans fraction) contained 3.4% proteins, 1.74% ash, and 88.3% dietary fiber and was used for the experiments without further pre-treatment [26]. The extract was named Gledi.

### 2.3. Gluten-Free Breadmaking

Gluten-free bread was formulated according to Sciarini et al. [30] with some modifications. The formulation included 45 g of rice flour, 45 g of cassava starch, 10 g of soy flour, 2 g of salt, 2 g of shortening, 3 g of compressed yeast and 80 g of water. Two levels of hydrocolloids were included, 0.5 and 1.25% (flour basis), according to previous studies [5,31]. Table 1 shows the amounts of hydrocolloids used in each formulation. When hydrocolloid mixtures (Gledi-Xanthan and Guar-Xanthan) were used, they were mixed in a 1:1 ratio. Consequently, eight gluten-free bread samples (those that include hydrocolloids) and the control sample (without hydrocolloids) were analyzed.

The ingredients were put together and mixed using a planetary mixer (Peabody SmartChef, Buenos Aires, Argentina) for 2 min at high speed. Hydrocolloids were dispersed in water under stirring for 30 min at room temperature to facilitate their dispersion previously to put together the rest of the ingredients. The batter was proofed for 30 min (30 °C and 85% relative humidity), and then, it was mixed again for 1 min at low speed in order to redistribute air cells and nutrients to improve yeast’s activity and to increase air incorporation into the batter. Afterward, the batter was weighed into aluminum cups (60 g), proofed again under the same conditions (30 min, 30 °C, and 85% relative humidity), and finally, baked at 180 °C for 30 min in a forced convection oven (Pauna-Cst, Buenos Aires, Argentina). The baked loaves were cooled at room temperature for 2 h and then were stored in polyethylene bags at 25 ± 2 °C until analysis. Breadmaking was performed in duplicate.

### 2.4. Batter Evaluation

#### 2.4.1. Batter Viscoelastic Properties

The viscoelastic behavior of the batters was analyzed using a controlled-stress rheometer (MCR 301, Anton Paar, Graz, Austria) equipped with a Peltier temperature controller. For these assays, batters were prepared according to Section 2.3, but without yeast addition and evaluated 30 min after their preparation. A strain sweep test was performed in the amplitude range of 0.01–100% at 1 Hz and 25 °C using a plate-plate configuration (50 mm ø, gap size 1 mm) to determine the linear viscoelastic range (LVR). Frequency sweep tests were made in the 0.01–100 Hz range at the strain 0.05% (LVR) with a plate-plate geometry (50 mm ø, gap size 1 mm). Elastic (G′) and viscous (G′′) moduli and complex viscosity (η*) were analyzed as a function of frequency. Batter preparation was performed in duplicate and analyzed.

#### 2.4.2. Batter Texture

Batters fractionated (30 g) into plastic containers (height: 45 mm; diameter: 45 mm) were analyzed immediately after mixing (Section 2.3) and proofing (60 min, 30 °C, 85% relative humidity) by a penetration test using an INSTRON texture analyzer (Universal Testing Machine, model 3342, Massachusetts, USA). Penetration test was made by a flat-faced metallic cylindrical probe (25 mm diameter). Test parameters were: load cell, 500 N, trigger force, 0.05 N, test speed, 5 mm/s, and 28 mm penetration. Tests were carried out at room temperature (25 ± 2 °C). Penetration load vs. penetration distance curves were recorded, and maximum force (batter strength (FM) and Young’s Modulus (resistance to elastic deformation (YM)) were determined. Batter preparation was performed in duplicate, and three determinations were performed in each batter batch.

### 2.5. Bread Quality

#### 2.5.1. Specific Bread Volume (SBV)

The volume of each bread loaf was determined by rapeseed displacement. Specific volume was obtained by dividing bread volume/bread weight. Specific bread volume was evaluated 2 h after baking. Four measurements of each breadmaking replication were performed.

#### 2.5.2. Bread Crust Color 

The crust color of bread was determined according to AACC method 14–22 [32] using a portable reflectance spectrophotometer (CM-700d/600d Konica Minolta, Ramsey, NJ, USA). Color parameters (L* = lightness, a* = position between red and green, and b* = position between yellow and blue) were recorded as CIE-LAB. Total color change (ΔE) was calculated as ΔE = (Δa*^2^ + Δb*^2^ + ΔL*^2^)^1/2^. Crust color was evaluated at 2 h after baking. Four measurements of each breadmaking replication were performed.

#### 2.5.3. Crumb Structure

Crumb structure was evaluated by image analysis using software Image J v1.45 s (National Institutes Health, Bethesda, MD, USA). For each bread piece, two slices were obtained, which were scanned (HP Scanjet G3010, Palo Alto, CA, USA). Digital images were binarized, according to Ribotta et al. [33]. Cell average area (mm^2^) and the number of cells/mm^2^ were determined. Crumb images were taken 2 h after baking. Four slices from two different bread loaves were analyzed in each breadmaking batch.

#### 2.5.4. Crumb Texture

Crumb texture was evaluated by texture profile analysis (TPA) test using an INSTRON texture analyzer (Universal Testing Machine, model 3342, Norwood, MA, USA). Bread slices were compressed using a flat-faced metallic cylindrical probe (25 mm diameter). Test parameters were: loadcell, 500 N; trigger force, 0.05 N; 40% compression. Tests were performed at room temperature (25 ± 2 °C). Crumb firmness (peak force during the first compression cycle) and chewiness were determined. Crumb texture was evaluated at 2, 24 and 72 h after baking. Four slices from 2 different bread loaves were evaluated in each breadmaking batch. The bread was stored in sealed plastic bags at 25 ± 2 °C until analysis. 

#### 2.5.5. Crumb Water Activity

Crumb water activity was determined using a dew-point hygrometer (AquaLab Pre, Lleida, Spain) and was carried out at (25 ± 0.5 °C). Crumb water activity was evaluated at 2, 24 and 72 h after baking. Four slices from two different bread loaves were analyzed in each breadmaking batch.

### 2.6. Statistical Analysis

The data were statistically analyzed by variance analysis (ANOVA); the means were compared by LSD Fisher test at a significance level of 0.05. The relationship between measured parameters was assessed using the Pearson test (significant levels at *p* ≤ 0.05) using Infostat Statistical Software (InfoStat-version2011. InfoStat Group. Argentina) [34].

## 3. Results and Discussion

### 3.1. Batter Viscoelastic Properties

The viscoelastic profile of the gluten-free batters showed solid-like behavior (G′ higher than G′′ over the frequency range evaluated), and as a consequence of the moduli slight frequency-dependence (mainly at low frequencies), samples were associated with a pseudo-gel-like behavior (Figure 1). The incorporation of hydrocolloids increased the viscoelastic profile. Considering the elastic modulus at 1 Hz (Table 2), Gledi incorporation at 0.5% and 1.25% increased the elastic component (G′) by 200% and 1500% compared to the control, respectively. The increase in Gledi dose increased G′ values by 400%. Regarding the combination of Gledi-Xanthan gum, the incorporation of 0.5% (0.25–0.25) and 1.25% (0.63–0.63) increased by 580% and 1100% of the elastic component, respectively. The increase in Gledi-Xanthan gum dose increased G′ values by 80%. The viscoelastic behavior of the batter containing Gledi 0.5% was smaller than Gledi-Xanthan 0.5%, whereas the viscoelastic profile of Gledi 1.25% and Gledi-Xanthan 1.25% was similar. The shape of tan δ (G′′/G′) profiles was similar among samples; initially, the solid properties of the batters increased as frequency increased from 0.01 to 1 Hz, and then the solid characteristic remained mostly unchanged between 1–100 Hz. The incorporation of Gledi 1.25% and both combinations of Gledi-Xanthan reduced the tan δ at lower frequencies (0.01–1 Hz), indicating that the batters showed more solid-like properties as a result of these hydrocolloid additions. At 1 Hz, tan δ was uninfluenced by any hydrocolloid incorporations (Table 2); however, in the region between 1–100 Hz, the tan δ of the batters containing Gledi and Gledi-Xanthan 1.25% was higher than the other samples, indicating that their solid-like properties are lower. The complex viscosity (η*) profile was similar for all samples, and it decreased as frequency increased (Figure 1). The batters containing hydrocolloids were significantly more viscous than the control, being Gledi 1.25% and Gledi-Xanthan 1.25%, the batters with the highest viscosity values. The viscosity profile of batter with Gledi 0.5% was lower than Gledi-Xanthan 0.5%, and there were no significant differences between the viscosity profile of the Gledi 1.25% and Gledi-Xanthan 1.25%.

Based on the batter formulation, the microstructure of batter can be assumed to be mostly made up of solids as a dispersed phase consisting of starch granules (representing about 50% of the system) and globular protein aggregates (representing about 6% of the system), suspended in a continuous aqueous phase (representing about 44% of the system). The high water proportion in gluten-free formulations is necessary to promote a certain viscosity of the batters to control the gas-holding capacity of the systems during proofing and to allow the complete disruption of starch granules during the gelatinization that takes place during baking, which is important for the formation of a cohesive structure during cooking and cooling process of the bread loaves. When hydrocolloids are included, they mostly form part of the continuous phase as a dispersed macromolecule, although they could also form a hydrocolloid-rich phase, depending on their solubility, making batters more viscous mainly because of their water-binding capacity. Considering Gledi galactomannan, it can be said that it contributes to increasing the system viscoelasticity and viscosity by water immobilization (similarly to other hydrocolloids) as a consequence of a higher continuous phase viscosity, which indirectly favors the particle-particle (rigid starch granules) interactions (particle crowding) of the dispersed phase. As mentioned before, the synergistic interaction between Xanthan gum and galactomannans, such as Guar and Locust bean gums, has been described. Likewise, the synergistic effect between the galactomannan extracted from Gt seeds from Portugal and Xanthan gum has been reported by Pinheiro et al. [21]. These authors informed that for Gledi-Xanthan systems (hot mixing: 30 min at 80 °C), the maximum synergy occurs for an 80:20 ratio, and this synergistic effect is reduced gradually as the galactomannan and Xanthan gum amounts decreased and/or increased proportionally.

According to the tan δ (G′′/G′) values, the microstructure of the batter was altered because of Gledi incorporation, and its effect is possibly due to an increase in solids effective concentration and/or a change in the plasticizing properties of water in the system. The microstructure arrangement of the gluten-free batter could result mainly from the interactions between the starch granules, the starch granules/globular-protein aggregates association through hydrogen bonds, and protein self-association through hydrogen bonds and electrostatic interaction. Taking this into account, probably, the more solid-like properties of batter containing Gledi 1.25% and both combinations of Gledi-Xanthan at low frequencies region (tan δ at 0.01–0.1 Hz) could be the result of a more interconnected internal structure through the hydrocolloid, where the components of each phase (dispersed and continuous) might establish interactions, e.g., continuous phase: protein/hydrocolloids, and hydrocolloid self-association through hydrogen bonds; dispersed phase: starch granules/hydrocolloid-rich phase and globular-protein aggregates/hydrocolloid-rich phase interactions. On the other hand, the microstructure of the batters with the highest hydrocolloid concentrations became less solid-like at high frequencies (tan δ at 1–100 Hz), which could be related to the weakening of the particle crowding as a result of the higher viscosity of the continuous phase which hinders the movements of particles/macromolecules. Additionally, it is possible that the interactions which contribute to making the microstructure more solid-like may be the result of transient associations more than stable interactions.

The viscoelastic behaviors of the batters with Gledi and Gledi-Xanthan addition were compared to that of Guar gum in order to compare the potential of Gledi galactomannan as a food hydrocolloid. The effect of the incorporation of Guar gum into the gluten-free batter was in agreement with previous studies [11]. The solid-like behavior of the batters containing Guar gum was similar to that of the batters containing Gledi (Appendix A). However, Guar seemed to bind more water than Gledi under the studied conditions, further increasing the batter’s viscoelastic behavior and viscosity. As observed in batters containing Gledi, batter with Guar 1.25% and Guar-Xanthan 1.25% presented the highest viscoelastic profiles without differences between them, and the viscoelastic behavior of the batter containing Guar 0.5% was smaller than Guar-Xanthan 0.5%. Regarding the elastic modulus and complex viscosity, Guar gum at 0.5% and 1.25% led to an increase two- and seven-fold higher than the increase caused by Gledi at these concentrations, respectively (Appendix A). This result might be related to the lower water solubility of Gledi, taking it longer to disperse. In this regard, Gledi and Guar dispersion kinetic at 1% *w*/*v* was evaluated by viscosity changes over time (20 rpm; 70 min). The dispersion process was described using a first-order kinetic [35]. The time required to reach 80% of Gledi and Guar dispersion was 98 and 21 min at 25 °C, respectively. In addition, the lower viscosity of Gledi systems compared to Guar systems could be associated with a more entangled conformation of the polysaccharide. It seems that, in an aqueous solution, the polymer segments interact with each other more than with the surrounding water molecules. The polymer then assumes a compact-like configuration reducing its binding water capacity and leading to lower viscosities.

The combination of Guar with Xanthan gum at 0.5% and 1.25% increased the elastic modulus and complex viscosity three and 10 times more than Gledi at the same concentrations, respectively (Appendix A). It has been informed that the synergistic effect in galactomannan/Xanthan systems depends on the mannose:galactose (Man:Gal) ratio as well as the fine structure of the galactomannans. In regard to the interaction mechanism of these polysaccharides, it is mostly accepted that the unsubstituted regions of galactomannans can establish interactions with disordered segments or chains of Xanthan molecules. The galactomannan from Gt, like other galactomannans, is a neutral heteropolysaccharide composed of a β-(1→4)-D-mannopyranosyl backbone substituted to varying degrees at α-(1→6) with single D-galactopyranosyl residues. Considering that the Man:Gal ratio of Gledi is higher (2.5:1) than Guar gum (2:1) [23], Gledi galactomannan should have more unsubstituted regions than Guar gum and consequently establish more interactions regions with Xanthan molecules. However, this does not seem to happen, probably as a consequence of the lower water-solubility of Gledi compared to Guar and the fact that the extension and occurrence frequency of the unsubstituted regions in the galactomannan chain is influenced by the galactose distribution pattern on the mannan backbone [29].

### 3.2. Batter Texture

The batters’ texture was evaluated to obtain further information about the influence of Gledi galactomannan incorporation in these systems. The changes in the batter properties were analyzed before and after proofing, and the results are presented in Table 3. As a result of the fermentation process, batter without hydrocolloids showed higher elastic resistance (Young’s modulus); however, the batter firmness was uninfluenced by this process. The incorporation of hydrocolloids mostly increased batter firmness and elastic resistance before and after proofing, although the degree of this increase was lower for the proofed batters. Gledi incorporation at 0.5% increased the elastic resistance (by 118%) of the unfermented batter; however, the texture of the fermented batter was similar to the fermented control sample. Gledi incorporation at 1.25% increased the batter firmness (400%) and the elastic resistance (300%) of the batter prior to the fermentation process, whereas the fermented batter showed similar texture parameters to the fermented control sample. Concerning the combination of Gledi-Xanthan gum, the incorporation of 0.5% and 1.25% increased the batter firmness before (300% and 880%, respectively) and after (125% and 400%, respectively) proofing. The same trend was observed for the elastic resistance. Comparing the textural properties of the unfermented and fermented batters, it is observed that batters containing Gledi had lower values than batters containing Gledi-Xanthan, with Gledi-Xanthan samples showing the highest ones. Thus, a synergistic effect between Gledi galactomannan and Xanthan gum can also be hypothesized when considering the texture properties of batters.

The gas retention capacity and tolerance to the fermentation process depend on batter bulk rheological and interfacial properties. During the proofing process, carbon dioxide becomes part of the system as a dispersed phase. The dispersed bubbles in the continuous phase are mainly surrounded by starch granules, globular protein aggregates, and the rest of the components dispersed in the continuous phase. The starch granules, as solid particles, can contribute to avoiding the coalescence of gas bubbles, behaving like a mechanical barrier that helps to reduce the interfacial tension between the bubble-aqueous phase [36]. Hydrocolloids increase the viscosity of the system, helping to further stabilize the interface between gas bubbles and the aqueous phase, which prevents the cell-gas coalescence and favors the batter expansion [5]. 

Both Gledi doses contributed to making the batter more elastically resistant and firmer before the proofing process, as shown by the rheological behavior of the macrostructure and the viscoelastic properties of the microstructure of these systems. However, the rheological properties of the macrostructure of fermented batters were similar to the fermented control sample, indicating that the changes in elastic resistance and firmness were not enough to modify the batter texture properties after the proofing process. On the other hand, the combination of Gledi-Xanthan gum resulted in more elastically resistant and firmer batters both before and after proofing. The texture properties of the unfermented batters agreed with the viscoelastic properties of their microstructure; the elastic resistance and firmness of fermented batters suggest a higher proportion of trapped gasses, indicating that Gledi-Xanthan gum would contribute to stabilizing the interface between the bubble gas and continuous aqueous phases. The Gledi-Xanthan gum combination increases the system’s viscosity by binding water, favoring the starch granules’ packing and, consequently, their Pickering stabilization effect on the bubbles gas [36]. However, the higher elastic resistance may also be related to a more resistant system to expand during proofing and baking, resulting in a compact breadcrumb at the end of the process.

The texture properties of the batters with Gledi and Gledi-Xanthan were compared to those containing Guar gum and Guar-Xanthan gum (Appendix A). The rheological profile of the batters containing Gledi was similar to that of batters containing Guar gum. Guar gum increased to a greater extent the batter firmness (1.5 and 1.3 times more for 0.5 and 1.25% addition, respectively) of the unfermented systems compared to Gledi. Unlike Gledi, Guar gum made the fermented batters firmer at both concentrations. Regarding the effects of the hydrocolloid combination, there were no differences in the batter firmness (unfermented and fermented) containing Gledi-Xanthan gum and Guar-Xanthan gum. Accordingly, from a rheological standpoint, Gledi galactomannan appears to be a suitable alternative to be used as a food hydrocolloid similar to Guar gum.

### 3.3. Bread Quality

The technological characteristics of bread with and without hydrocolloids were assessed by measuring their specific volume, crumb structure, and crumb water activity. In addition, the evolution of crumb firmness and chewiness during 3 days of storage at room temperature was recorded. Gledi addition, both at 0.5% and 1.25%, significantly increased bread volume compared to the control, whereas when Xanthan gum was included, a decrease was observed, especially at higher doses (Table 4). 

As shown in Figure 2 and Table 4, when Gledi was added, the crumb structure was more open and seemed more uniform than the other samples. The increase in SBV was accompanied by a decrease in initial crumb firmness and chewiness, and during storage, the increases in firmness and chewiness were greatly reduced by hydrocolloid addition. The same trend was observed for a_w_, although slight differences were found; a_w_ was more stable during storage in samples with hydrocolloids (Table 5 and Table 6). As expected, the effect of hydrocolloids on water distribution resulted in changes in the Maillard reactions during crust formation, leading to a lighter (higher L* values), reddish (higher a*) and yellowish (higher b*) crust, in agreement with previous findings [37,38] (Table 7). These changes resulted, as expected, in overall ΔE changes. When Guar gum was used, the same trend was observed, although changes were more pronounced (Appendix A).

Types of bread prepared using Guar gum and Guar-Xanthan gums combinations were also analyzed, and the observed trend was comparable to that of bread with Gledi and Gledi-Xanthan gum (Appendix A and Appendix A).

During baking, a series of changes occur in the batters, mainly governed by starch gelatinization and protein denaturation and coagulation. During starch gelatinization, part of the amylose is leached out of the granule and becomes exposed and available to interact with different hydrophilic macromolecules, mainly proteins and hydrocolloids. The interaction between starch (mainly from cassava) and proteins (mainly from active soy flour) has already been reported elsewhere [30]. Moreover, it has been observed that, upon gelatinization, cassava starch loses its granular structure, presenting a homogeneous network-like structure, compared to other starches that display a continuous phase formed by swollen starch granules tightly interacting [39]. 

In this context, it is worth highlighting that Gledi has a slow solubility at room temperature, which reduces its initial water absorption capacity and leads to a more gradual viscosity increase of the batter. On the one hand, this may result in higher water availability for starch gelatinization, which in turn results in a stabilization of the system structure, allowing gasses trapped during fermentation to expand during baking. On the other hand, the delayed viscosity increase may lead to greater volume expansion during proofing when the batter is probably less firm. These changes stabilize the structure of the system, allowing gasses trapped during fermentation to expand during baking. Likewise, the increased viscoelasticity of the batters produced by Gledi would also play a stabilizing role in the system. 

The presence of Xanthan gum, in all cases, led to a decrease in bread quality. Thus, although the micro- and macro-structure of the batters point to the formation of more stable systems, it is likely that their higher binding water capacity inhibits, to a certain degree, the swelling and gelatinization of the starch granules, decreasing the quality of the bread obtained.

## 4. Conclusions

The results presented herein indicate that the galactomannan extracted from *Gleditsia triacanthos* seed modifies both the micro- and macrostructure of gluten-free batters, which has an impact on the rheological and textural properties of the system. These changes also influence the final quality of the bread.

The incorporation of Gledi increased the viscoelasticity, firmness, and elastic resistance of the batters. These parameters were even higher when combined with Xanthan gum; however, when considering the quality of the bread, the incorporation of Xanthan gum had a negative effect, indicating that there is an optimal rheological and textural behavior of the batters, beyond which the quality of the final product decreases.

Finally, the performance of Gledi was similar to that of Guar gum, used as a reference galactomannan, but with a lower solubility at room temperature, which would favor the production of bread of better technological quality, i.e., higher specific volume, lower crumb firmness and firming rate during storage.

Based on these results, the sensory evaluation of different gluten-free bread formulations with the incorporation of Gledi is proposed as a future perspective. On the other hand, we intend to work for the inclusion of Gt galactomannan in the list of permitted additives for food use.

## Figures and Tables

**Figure 1 foods-12-00756-f001:**
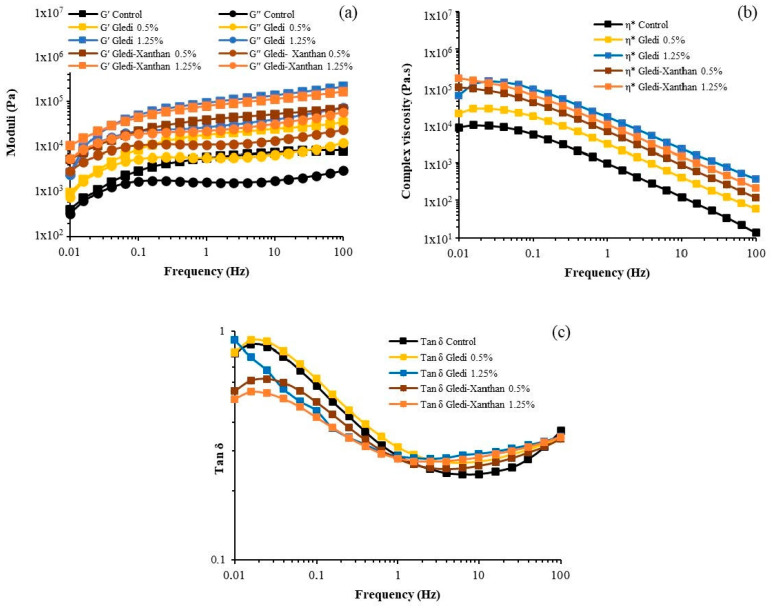
Rheograms for all studied samples. (**a**) Storage and loss moduli, (**b**) Complex viscosity, (**c**) tan δ.

**Figure 2 foods-12-00756-f002:**
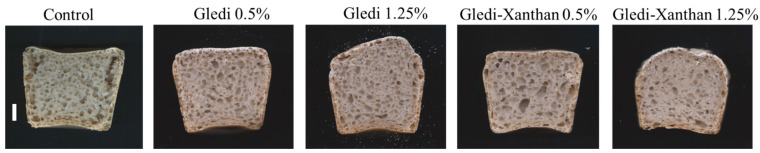
Representative images of gluten-free bread. Bar: 1 cm.

**Table 1 foods-12-00756-t001:** Hydrocolloids used for gluten-free breadmaking.

Hydrocolloids	g/100 g Flour-Starch
Gledi 0.5%	0.5
Gledi 1.25%	1.25
Gledi-Xanthan 0.5%	0.25–0.25
Gledi-Xanthan 1.25%	0.63–0.63
Guar 0.5%	0.5
Guar 0.5%	1.25
Guar-Xanthan 0.5%	0.25–0.25
Guar-Xanthan 1.25%	0.63–0.63

**Table 2 foods-12-00756-t002:** Rheological parameters of gluten-free batters.

Sample	Storage Modulus (Pa)	Loss Modulus (Pa)	tan δ	Complex Viscosity (Pa.s)
Control	5710 ± 183 a	1630 ± 70 a	0.286 ± 0.005 ab	945 ± 31 a
Gledi 0.5%	18,175 ± 3221 b	5655 ± 1050 b	0.297 ± 0.023 b	3025 ± 539 b
Gledi 1.25%	93,700 ± 21,871 de	26,500 ± 5789 de	0.284 ± 0.006 ab	15,525 ± 3629 de
Gledi-Xanthan 0.5%	39,300 ± 6206 c	10,915 ± 1861 c	0.278 ± 0.005 a	6495 ± 1027 c
Gledi-Xanthan 1.25%	70,766 ± 18,694 d	19,616 ± 5167 d	0.277 ± 0.006 a	11,686 ± 3093 d

Different letters within a column are significantly different (*p* < 0.05).

**Table 3 foods-12-00756-t003:** Textural parameters of gluten-free batters.

Sample	Unfermented Batter		Fermented Batter	
	Firmness (g)	Young’s Modulus (Pa)	Firmness (g)	Young’s Modulus (Pa)
Control	23 ± 4 a	2129 ± 694 a	21 ± 2 a	5923 ± 1692 a
Gledi 0.5%	41 ± 4 a	4635 ± 751 b	19 ± 2 a	6669 ± 1628 ab
Gledi 1.25%	114 ± 15 c	8525 ± 1677 c	22 ± 3 a	9282 ± 2020 abc
Gledi-Xanthan 0.5%	95 ± 8 b	8622 ± 735 c	48 ± 6 b	9714 ± 3461 bc
Gledi-Xanthan 1.25%	227 ± 28 d	15,615 ± 1945 d	106 ± 17 c	10,816 ± 3531 c

Different letters within a column are significantly different (*p* < 0.05).

**Table 4 foods-12-00756-t004:** Gluten-free bread’s specific volume and crumb structure.

Sample	Cell Area (%)	Cell/mm^2^	Cell Size (mm)	SBV (cm^3^/g)
Control	48.5 ± 0.3 a	0.65 ± 0.02 c	0.75 ± 0.02 a	1.99 ± 0.30 b
Gledi 0.5%	49.8 ± 0.6 b	0.35 ± 0.02 a	1.45 ± 0.04 c	2.24 ± 0.23 c
Gledi 1.25%	49.9 ± 0.0 b	0.34 ± 0.04 a	1.51 ± 0.19 c	2.23 ± 0.28 c
Gledi-Xanthan 0.5%	50.4 ± 0.0 b	0.35 ± 0.03 a	1.60 ± 0.04 c	1.96 ± 0.16 ab
Gledi-Xanthan 1.25%	50.3 ± 0.2 b	0.45 ± 0.05 b	1.05 ± 0.00 b	1.74 ± 0.09 a

SBV: specific bread volume. Different letters within a column are significantly different (*p* < 0.05).

**Table 5 foods-12-00756-t005:** Gluten-free bread textural parameters.

Sample	Firmness (N)			Staling Rate	Chewiness (ad)		
	Day 0	Day 1	Day 3	(N/day)	Day 0	Day 1	Day 3
Control	5.6 ± 1.2 c	16.1 ± 2.1 c	42.7 ± 8.9 b	12.51	6.7 ± 1.3 c	14.2 ± 1.5 d	23.6 ± 5.8 c
Gledi 0.5%	4.5 ± 0.8 ab	10.3 ± 1.1 b	24.6 ± 4.6 a	6.77	5.2 ± 0.6 a	9.2 ± 1.2 b	14.1 ± 2.8 b
Gledi 1.25%	3.8 ± 0.8 a	7.9 ± 1.4 a	19.9 ± 6.8 a	5.47	4.5 ± 0.9 a	7.5 ± 1.3 a	10.4 ± 2.5 a
Gledi-Xanthan 0.5%	4.4 ± 1.2 ab	9.5 ± 2.0 ab	21.3 ± 3.5 a	5.66	5.4 ± 1.4 ab	10.2 ± 1.4 bc	13.1 ± 2.9 ab
Gledi-Xanthan 1.25%	5.2 ± 0.5 bc	10.9 ± 3.0 b	23.8 ± 5.1 a	6.22	6.3 ± 0.4 bc	10.9 ± 2.2 c	16.5 ± 1.5 b

Different letters within a column are significantly different (*p* < 0.05).

**Table 6 foods-12-00756-t006:** Crumb water activity.

Sample	a_w_		
	Day 0	Day 1	Day 3
Control	0.977 ± 0.001 a	0.976 ± 0.001 a	0.967 ± 0.003 a
Gledi 0.5%	0.977 ± 0.001 a	0.976 ± 0.001 a	0.970 ± 0.002 a
Gledi 1.25%	0.977 ± 0.002 a	0.976 ± 0.001 a	0.970 ± 0.002 b
Gledi-Xanthan 0.5%	0.978 ± 0.002 a	0.976 ± 0.001 a	0.971 ± 0.002 b
Gledi-Xanthan 1.25%	0.978 ± 0.001 a	0.975 ± 0.001 a	0.972 ± 0.002 b

Different letters within a column are significantly different (*p* < 0.05).

**Table 7 foods-12-00756-t007:** Gluten-free bread crust color.

Sample	L*	a*	b*	ΔE
Control	72.2 ± 4.6 a	2.5 ± 0.4 a	20.3 ± 3.0 a	-
Gledi 0.5%	73.7 ± 1.1 ab	5.1 ± 1.2 b	26.5 ± 3.5 b	6.89
Gledi 1.25%	74.8 ± 1.0 ab	4.7 ± 0.9 b	26.0 ± 1.0 b	6.64
Gledi-Xanthan 0.5%	76.1 ± 0.8 b	4.4 ± 0.8 b	26.9 ± 1.8 b	7.89
Gledi-Xanthan 1.25%	76.0 ± 1.1 b	3.0 ± 0.7 a	22.4 ± 1.4 a	4.37

Different letters within a column are significantly different (*p* < 0.05). ΔE represents the total color change of samples with hydrocolloids compared to the control.

## Data Availability

The data are available from the corresponding author.

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
