# Peer review of "Gleditsia triacanthos Galactomannans in Gluten-Free Formulation: Batter Rheology and Bread Quality"

_foods, 2023, doi:10.3390/foods12040756_

Round 1

Reviewer 1 Report

This is an interesting and innovative paper reporting the potential use of Gledi galactomannans to improve gluten-free formulation of bread.

However, some points should be clarified by the authors.

Introduction:

-the market of glute-free products should be contextualized, especially regarding celiac patients, who need high quality formulations to improve the nutritional aspects and adherence to the diet.  

-Are Gleditsia triacanthos galactomannans already admitted as additives in south America or Europe? This aspect influences the possibility to expand the market of glute-free products.

Bread quality:

-Did the authors evaluated also variations in term of firmness and chewiness during the same day? The worst characteristic of gluten free bread is the early firmness that makes this food one of the most critical for celiac patients.

-A panel test on the different formulations, especially those containing Gledi should be performed. This aspect could allow to evaluate not only products organoleptic properties but also the perception of the different parameters reported (firmness, chewiness…).

In the conclusion section, authors should add the future perspective on the basis of the results obtained in the study.

Author Response

Thank you to both Reviewer 1 and 2 for the favorable comments. We have made changes in the manuscript according to their suggestions.

Also a point by point responses to the referees’ comments are below.

Reviewer 1

This is an interesting and innovative paper reporting the potential use of Gledi galactomannans to improve gluten-free formulation of bread.

However, some points should be clarified by the authors.

Introduction:

-the market of glute-free products should be contextualized, especially regarding celiac patients, who need high quality formulations to improve the nutritional aspects and adherence to the diet. 

The text was modified according to reviewer suggestion

“Celiac disease (CD) is an autoimmune disorder caused by the consumption of gluten proteins; the only treatment currently available for this disease is patient adherence to a lifelong gluten-free diet. The frequency of CD has been reported to double every two decades [1]. This, coupled with the perception of some non-celiac consumers that the gluten-free diet is healthier, has resulted in a large increase in the size of the market for these food products. Thus, some reports estimate the size of the gluten-free market to be USD 5.7 billion in 2020, and it is projected to reach USD 8.3 billion in 2025 [2]. This situation poses a challenge for the scientific community and the food industry. Finding new ingredients, and combinations of raw materials and production processes to obtain gluten-free food is necessary to fulfill consumers’ demand for this type of products. Among gluten-free products, the manufacture of quality gluten-free breads is one of the most challenging tasks, largely due to the lack of alternative ingredients that can mimic wheat protein functionality [3].”

This information was added in INTRODUCTION.

-Are Gleditsia triacanthos galactomannans already admitted as additives in south America or Europe? This aspect influences the possibility to expand the market of glute-free products.

The galactomannan from Gleditsia Triacanthos has not yet been admitted as an additive in our country, but there are some interested companies, so we believe that the conditions are right to initiate the procedures in the corresponding organisms in the near future.

Bread quality:

-Did the authors evaluated also variations in term of firmness and chewiness during the same day? The worst characteristic of gluten free bread is the early firmness that makes this food one of the most critical for celiac patients.

Crumb texture was evaluated at 2, 24 and 72 h after baking. As the reviewer mentions, the early firmness is critical. We believe that the determination 2, 24 and 72 h after baking allows a convenient evaluation of Gt galactomannan in combination with xanthan gum.

-A panel test on the different formulations, especially those containing Gledi should be performed. This aspect could allow to evaluate not only products organoleptic properties but also the perception of the different parameters reported (firmness, chewiness…).

We totally agree with the comment of the reviewer. This study aimed to evaluate the incorporation of Gledi galactomannan, alone and in combination with xanthan gum, in gluten-free batters and breads, and compared it with guar gum. However, next steps are to initiate the procedures to admit it as an additive in the corresponding organisms, and also test different formulations with sensorial panels.

In the conclusion section, authors should add the future perspective on the basis of the results obtained in the study.

The future perspectives were added in conclusions: “Based on these results, the sensory evaluation of different gluten-free bread formula-tions with the incorporation of Gledi is proposed as a future perspective. On the other hand, we intend to work for the inclusion of Gt galactomannan in the list of permitted ad-ditives for food use.”

Reviewer 2 Report

The authors present an innovative approach to test a new source of hydrocolloid for creating the texture of gluten-free bread. Previous publications by the authors have dealt with the characteristics of the plant and seed-derived galactomannans, so I consider the article original and bringing new knowledge also applied.

The introduction was written correctly - the authors presented the current state of knowledge about hydrocolloids used in gluten-free formulations. They briefly introduced the plant and at the end stated the purpose and scope of the work. The purpose was well justified.

line 54-55- Authors need to develop the abbreviation Man:Gal when they first use it.

In Introduction authors needs to include this research  about using Gleditsia triacanthos (Fabaceae) in creation of edible films, authors there proved the synergic effect between components including seed-derived galactomannans allowed the development of films with improved reology https://doi.org/10.1016/j.foodhyd.2019.105227

I have no comments on the methodology - it was given comprehensively so that it can be reproduced.

Figures or Tables should be placed closer to first mention in the text. Now they are grouped together by making it necessary to look for them far from the place of mention.

The authors very meticulously and thoroughly describe the dependence of the composition of the hydrocolloid mixture or Gledi extract alone on the rheological properties of the dough and then bread. The authors have chosen well the analyses to present the effect of Gledi extract on the texture and physicochemical properties of dough and bread.

In the case of the color test, the authors should calculate and provide the total color difference ΔE* in addition to or instead of L*, a*, b* values.

The authors have written conclusions that follow from their research and are original. However, they are very general and should be made more specific, especially since the authors have done a great deal of research.

Author Response

Thank you to both Reviewer 1 and 2 for the favorable comments. We have made changes in the manuscript according to their suggestions.

Also a point by point responses to the referees’ comments are below.

The authors present an innovative approach to test a new source of hydrocolloid for creating the texture of gluten-free bread. Previous publications by the authors have dealt with the characteristics of the plant and seed-derived galactomannans, so I consider the article original and bringing new knowledge also applied.

The introduction was written correctly - the authors presented the current state of knowledge about hydrocolloids used in gluten-free formulations. They briefly introduced the plant and at the end stated the purpose and scope of the work. The purpose was well justified.

line 54-55- Authors need to develop the abbreviation Man:Gal when they first use it.

The abbreviation was developed

In Introduction authors needs to include this research  about using Gleditsia triacanthos (Fabaceae) in creation of edible films, authors there proved the synergic effect between components including seed-derived galactomannans allowed the development of films with improved reology https://doi.org/10.1016/j.foodhyd.2019.105227

The research was included in Introduction (Ln 82-86): In a previous study, the improving effect on the film-forming properties of soy protein was reported when the hot-water soluble galactomannan extracted from Gt seeds was used as a co-component [22]. It was found that improved films could be obtained through the combination of both components in certain ratios in relation to single component samples.

I have no comments on the methodology - it was given comprehensively so that it can be reproduced.

Figures or Tables should be placed closer to first mention in the text. Now they are grouped together by making it necessary to look for them far from the place of mention.

Figures or Tables were moved according to reviewer suggestion

The authors very meticulously and thoroughly describe the dependence of the composition of the hydrocolloid mixture or Gledi extract alone on the rheological properties of the dough and then bread. The authors have chosen well the analyses to present the effect of Gledi extract on the texture and physicochemical properties of dough and bread.

In the case of the color test, the authors should calculate and provide the total color difference ΔE* in addition to or instead of L*, a*, b* values.

The calculations were added in tables 7 and S6

The authors have written conclusions that follow from their research and are original. However, they are very general and should be made more specific, especially since the authors have done a great deal of research.

Conclusions were modified according to the comment of the reviewer 1. In relation to the comment about to be more specific, since specific data was included in the abstract, we believe that also adding them in the conclusions would be repeating the information.

Round 2

Reviewer 1 Report

Considering that Gleditisia Triacanthos has not yet been ammitted as additive, I suggest the authors to include this information in the introduction or conclusion section, because the interest of companies and the promising results could enforce the impact of the study and support the request to organisms

Author Response

We have made the following incorporation in the manuscript (L87-92)

“The galactomannan extracted from Gt seeds with boiling water could be a potential ad-ditive in the food industry [23-24], and be used as an alternative to guar and locust bean gums [25]; moreover, synergistic interaction between this and other hydrocolloids has been reported [26-29]. However, it is important to mention that the extract of Gleditisia triacanthos has not been admitted yet as a food additive in Argentine legislation.”
